# QuantGen: Parameter Generation for Controllable Model Quantization

## Abstract

Parameter generation has recently emerged as a novel and effective paradigm in the pursuit of efficient AI, offering a fundamentally different perspective from conventional deep learning by directly synthesizing high-quality model parameters. Despite its promise, existing parameter generation methods are typically constrained to producing parameters aligned with the task objectives present in their training data. This limitation significantly restricts their applicability and practical utility across diverse real-world scenarios. In this work, we introduce a dedicated parameter generation method specifically designed for model quantization—a critical step in deploying deep learning models on resource-constrained devices. We propose the first recurrent-based generator capable of directly producing quantized model parameters that retain performance comparable to their full-precision counterparts, without requiring any additional data or retraining. Furthermore, our framework supports controllable quantization, enabling the generation of parameters that satisfy varying precision and deployment requirements. Extensive experiments across multiple datasets and model architectures demonstrate that our method achieves strong generalization and robustness under a wide range of quantization settings. These findings underscore the potential of parameter generation as a powerful and flexible tool for efficient model compression, training, and deployment.

## 1 Introduction

With the rapid advancement of deep learning, the use of increasingly large datasets and models has become a common strategy for achieving state-of-the-art performance across a wide range of tasks (Liu et al., 2024; Achiam et al., 2023; Zhao et al., 2024; Lin et al., 2023; Liu et al., 2023; Zhang et al., 2024). This trend has led to growing interest in efficient AI, which aims to accelerate the deployment of deep learning models while preserving their task effectiveness. Within this context, parameter generation has recently emerged as a promising and efficient paradigm, providing a compelling alternative to conventional training by enabling the direct synthesis of high-quality model parameters.

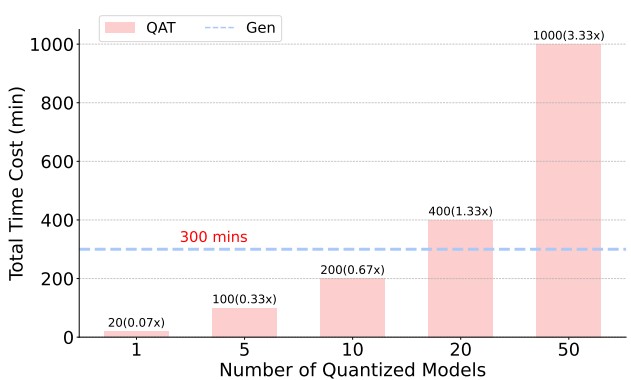

Figure 1: The total time cost of quantization using QAT and our quantized parameter generation model. As the inference time of ours is less one second, the overall time consumption remains effectively constant regardless of the number of target models.

Parameter generation conceptualizes model parameters as a novel modality and seeks to rapidly synthesize new parameters that adhere to the distributional patterns observed in a training set of pretrained models. Traditional approaches to parameter generation often rely on modeling the

probability distribution of parameters using Stochastic neural networks (Sompolinsky et al., 1988; Bottou, 1991; Graves, 2011) and Bayesian neural networks (Neal, 1995; Kingma et al., 2013; Gal & Ghahramani, 2015). However, such methods typically suffer from high computational complexity and limited generalization across diverse model architectures. Subsequent advancements, such as HyperNetworks (Ha et al., 2016) and SMASH (Brock et al., 2017), extended the applicability of parameter generation by enabling the generation of weights for other networks. Nevertheless, these methods still face scalability challenges due to the increasing size of modern deep learning models, thereby limiting their practical deployment. With the rapid development of generative AI (Brock et al., 2018; Song et al., 2020; Li et al., 2022; Dhariwal & Nichol, 2021; Peebles & Xie, 2023; Li et al., 2024), recent studies have employed diffusion models Song et al. (2020) to achieve substantial progress in parameter generation. For instance, P-Diff (Wang et al., 2024) demonstrated the capability to generate model parameters that achieve performance comparable to their original counterparts on classification tasks. Building on this foundation, Cond P-Diff (Jin et al., 2024) and Tina (Li et al., 2024) introduced text-conditioned parameter generation, offering enhanced flexibility and control. To further scale the generation capacity, RPG (Wang et al., 2025) and ORAL (Khan et al., 2025) proposed recurrent diffusion architectures which consists of recurrent model and diffusion to generate model parameters at the scale of hundreds of millions, and validated their effectiveness across a variety of tasks and architectures.

Despite recent advancements in parameter generation, most existing methods introduce substantial training and inference overhead due to the reliance on diffusion models, and are constrained to generating model parameters that strictly adhere to the task objectives of input parameters. This constraint significantly hinders the practical applicability of parameter generation in real-world deployment scenarios. To overcome these limitations, we pioneer the application of parameter generation in the practical domain of model quantization (Gholami et al., 2022; Yang et al., 2019; Polino et al., 2018; Gray & Neuhoff, 1998), thereby advancing the broader objective of efficient AI. Specifically, we eliminate the diffusion model used in previous methods (Wang et al., 2025; Khan et al., 2025) and propose a recurrent-based framework that enables one-step generation of quantized model parameters directly from their full-precision counterparts, without requiring access to the target task dataset or additional fine-tuning. Our method generates quantized parameters with performance comparable to that of traditionally quantized models. Furthermore, it supports a variety of quantization schemes and target precisions, enabling a conditionally controllable and flexible quantization process. Extensive experiments across diverse tasks and model architectures demonstrate that our approach achieves truly general-purpose and data-free model quantization (Nagel et al., 2019). Beyond efficient model compression, it also alleviates privacy and security risks associated with accessing sensitive training data in conventional quantization pipelines. Moreover, our work introduces a novel perspective on integrating parameter generation with real-world deployment challenges. Our main contributions are summarized as follows:

- This work presents the first application of parameter generation to model quantization, extending its utility beyond repeated parameter synthesis and addressing a practical, high-impact deployment challenge.

- We propose an efficient framework that eliminates the diffusion model, enabling one-step generation of quantized model parameters with controllable precision and quantization type, without requiring additional training or calibration data.

- Extensive experiments demonstrate strong generalization, competitive performance, and robustness across a wide range of tasks and architectures, offering a practical, privacy-preserving solution for real-world model quantization.

## 2 RELATED WORKS

### 2.1 PARAMETER GENERATION

The core idea of parameter generation lies in learning the distribution of existing model parameters to synthesize new ones that achieve performance comparable to the original models. Early studies approached this by modeling parameter distributions using Stochastic neural networks (Sompolinsky et al., 1988; Bottou, 1991; Graves, 2011) and Bayesian neural networks (Neal, 1995; Kingma et al., 2013; Gal & Ghahramani, 2015). However, these methods struggled with the high dimensionality of

modern model parameters, often failing to generate complete weight sets and exhibiting poor generalization across architectures. To overcome these limitations, approaches such as HyperNetworks (Ha et al., 2016) and SMASH (Brock et al., 2017) introduced auxiliary lightweight neural networks to generate parameters for various architectures, significantly enhancing the applicability of parameter generation.

The emergence of diffusion models has given rise to a new research direction that treats model parameters as a distinct modality and learns their distributions via diffusion-based frameworks. For instance, Hyper-Representations (Schürholt et al., 2022) employed an autoencoder to capture the latent distribution of pretrained model parameters. P-Diff (Wang et al., 2024) was the first to demonstrate the feasibility of directly generating parameters through diffusion models, successfully synthesizing batch normalization parameters for classification tasks. Building on this, Cond P-Diff (Jin et al., 2024) and Tina (Li et al., 2024) incorporated textual conditioning to enable controllable parameter generation.

Despite these advances, current methods remain limited to generating small-scale parameters and are unable to synthesize full parameter sets for large architectures such as ResNet, ConvNeXt, or ViT. To address this challenge, RPG (Wang et al., 2025) introduced a recurrent diffusion architecture that combines Mamba (Gu & Dao, 2023) and diffusion models, leveraging symbolic representations to scale up to hundreds of millions of parameters. ORAL (Khan et al., 2025) further extended this framework by introducing conditional control mechanisms into the generation process. However, these methods introduce significant training and inference overhead due to the use of diffusion models, and restricted to repeatedly reproducing parameters that follow the same task objectives of the input parameters. This constraint substantially limits the practical scope and versatility of parameter generation techniques.

## 2.2 MODEL QUANTIZATION

Model quantization is a fundamental technique for enabling efficient deployment of deep learning models on resource-constrained devices. It reduces memory consumption and computational overhead by converting floating-point weights and activations into lower-precision formats, such as 8-bit or even sub-4-bit integers. This procedure consists of two main stages: quantization and dequantization. The quantization process is represented by the following formula:

$$q = \text{round}\left(\frac{r}{S}\right) + Z, S = \frac{\max(|r|)}{2^{N-1} - 1}, \tag{1}$$

where $r$ represents input numbers, $q$ represents the quantized integers with $N$ bits, $S$ is the scaling factor, and $Z$ is an integer zero point. To recover the floating-point approximation, the quantized value is transformed using the equation:

$$\hat{r} = S(q - Z), \tag{2}$$

which is known as the dequantization. However, the recovered $\hat{r}$ will not be exactly equal to the original input $r$, due to the rounding step introducing an unavoidable approximation.

Post-training quantization (PTQ) methods (Nagel et al., 2020; Banner et al., 2019) aim to quantize models without retraining by analyzing the statistical properties of activation distributions. While PTQ methods are simple and efficient, they often suffer from substantial performance degradation at lower bit-widths or in complex architectures such as Transformers, where quantization sensitivity is heightened. To address these limitations, quantization-aware training (QAT) (Jacob et al., 2018; Esser et al., 2019) integrates quantization operations into the training loop, enabling the model to learn robustness to quantization noise. QAT approaches typically simulate quantization effects during both the forward and backward passes . Although QAT achieves superior performance and supports ultra-low-bit quantization, it incurs substantial training costs and typically requires access to large-scale labeled datasets.

In parallel, data-free quantization (Nagel et al., 2019; Li et al., 2021) has emerged as a promising direction for settings where access to training data is unavailable due to privacy concern. These approaches aim to synthesize representative inputs or directly estimate quantization sensitivity without relying on original samples. While advantageous in privacy-sensitive scenarios, these methods often involve brittle synthesis processes, increased computational complexity, and limited generalization under aggressive compression settings. Despite significant progress has been made in

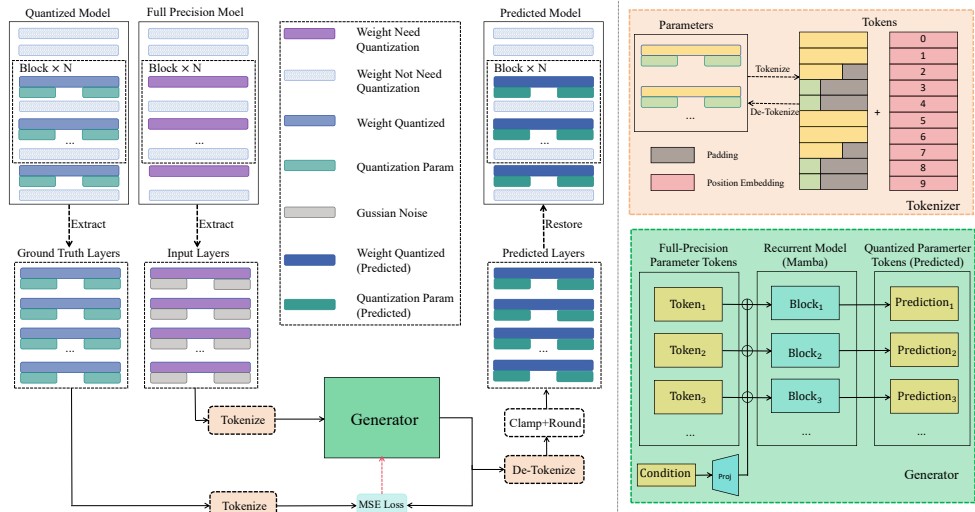

Figure 2: Overview of the proposed framework. The left part illustrates the construction and preprocessing of the training dataset for the parameter generation model, along with its training and inference pipeline. The top-right part shows the mapping between model parameters and the parameter tokens used by the generation model. The bottom-right part presents the internal architecture of the generation model and the integration of controllable condition embeddings.

balancing compression and performance, most existing quantization techniques still rely on either full supervision or handcrafted heuristics. These limitations highlight the need for more generalizable, efficient, and scalable quantization frameworks.

## 3 METHOD

In this section, we present the application of parameter generation to model quantization and outline the specific process of our proposed method. We first treat the full-precision parameters and their corresponding quantized parameters as training data. Then, we train a novel Mamba-based quantized parameter generation framework. Finally, during the sampling phase, we perform direct sampling of quantized parameters without requiring any additional training. The detailed pipeline and model of our proposed method are shown in Figure 2.

### 3.1 PARAMETER PROCESSING

Our objective is to generate quantized model parameters tailored to various tasks, model architectures, and quantization standards. To accomplish this, we first construct a training dataset consisting of the full-precision parameters and their corresponding quantized counterparts. Taking into account the unique characteristics of the quantization task and the specific requirements of recurrent models, we preprocess the parameters through the following steps to facilitate dataset construction.

**Quantization Information Alignment.** We begin by performing quantization on the existing full-precision parameters using various quantization precisions. In traditional quantization procedures, the affine transformation used for quantization inherently computes the scaling factor $S$ and zero point $Z$ based on the full-precision parameters and the target quantization range. However, since our method directly generates entirely new quantized parameters, the full-precision parameters used as input do not adhere to the same $S$ and $Z$ as those generated. Consequently, we incorporate the scaling factor $S$ and zero point $Z$ from the quantization process into the quantized parameter weights for the parameter generation model to learn. To maintain consistency in the model architecture, we insert fictitious $S$ and $Z$ values, initialized with Gaussian noise, at the corresponding positions in the original model.

**Parameter Tokenization.** To generate models with a large number of parameters, we adopt a tokenization strategy inspired by RPG (Wang et al., 2025), while introducing key distinctions tailored

to our framework shown in Figure 2. Before tokenizing the parameters, we first fill any blank positions pre-calculated in the final token with a constant value. We then flatten and normalize the quantized layer weights for both the original full-precision parameters and their corresponding quantized counterparts at the layer level, thereby maintaining the hierarchical structure of the parameters and enhancing training stability. Next, based on a predefined token length, we partition the normalized weights of each layer into tokens. Subsequently, we insert the scaling factor $S$ and zero point $Z$ into the two tokens that follow the corresponding layers, applying the same padding strategy to any vacant positions. This process is repeated until all layers are tokenized, and the tokens corresponding to various precisions are aggregated into a training dataset.

## 3.2 RECURRENT GENERATOR TRAINING

After processing the model parameters, we utilize the trainable recurrent model for parameter generation. To preserve the inter-layer and intra-layer relationships between different parameters that are split into tokens, we encode all tokens using an additional position embedding to identify different tokens and add it to the full-precision parameter tokens to form $\{t_1, \cdots, t_n\}$ as input to the Mamba model $f(\cdot)$. The recurrent mechanism captures the dependencies between tokens, which can be formulated as:

$$p_i, h_i = f(t_i, h_{i-1}), \tag{3}$$

where $p_i$ denotes the predicted quantized token, while $h_i$ denotes the hidden state. Unlike existing parameter generation methods (Wang et al., 2025; Khan et al., 2025), we do not pass the output sequence $\{p_1, \cdots, p_n\}$ into a diffusion model; instead, we directly calculate the MSE loss.

In the context of model quantization, what is required is a deep neural network that captures the affine transformation relationship between full-precision parameters and their corresponding quantized parameters, rather than learning complex distribution mappings. Therefore, the recurrent model alone is sufficient to satisfy this mathematical modeling requirement. Thus we calculate the difference between the predicted parameter tokens and the parameter tokens quantized by PTQ or QAT. The training objective can be expressed as:

$$L = \sum_{i=1}^{n} \|p_i - q_i\|^2, \tag{4}$$

where $q_i$ represents the $i$-th quantized parameter tokens within the ground truth set $\{q_1, \cdots, q_n\}$. Experimental results demonstrate that removing the diffusion model significantly enhances the performance of generated quantized model parameters, thereby confirming our hypothesis and facilitating the training and inference efficiency.

**Controllable Condition Guidance.** Building upon the above framework, we further investigate how to enable conditional control within the parameter generation model to produce quantized parameters that meet varying requirements. Specifically, for different quantization precisions or granularities, we introduce an additional one-hot encoded condition vector, with its dimensionality corresponding to the number of distinct quantization configurations. During both training and inference, this one-hot vector is transformed via a learnable linear projection into an embedding that aligns with the dimensionality of the full-precision parameter tokens and quantization information tokens. The resulting embedding is then added to the parameter tokens, along with positional embeddings, to form the final input to the generation model.

## 3.3 QUANTIZED PARAMETER INFERENCE

When the trained quantized parameter generation model performs a single quantization operation on the full-precision parameters to be quantized, since the training process is entirely based on fp-32 precision, the resulting quantized parameters must undergo rounding and clamping operations to ensure their values lie within the corresponding quantization precision range. Next, based on the token encoding, the flattened parameters are concatenated to form the original weights, which are then assigned to the corresponding quantization layer. During this process, we remove the scaling factor $S$ and zero point $Z$, preserving them for use in the testing procedure. At this point, we obtain the fully quantized model parameters.

Table 1: Comparison of PTQ and our quantized parameter generation model across datasets and architectures. The proposed method achieves consistently better performance, validating its robustness and general applicability.

| Dataset | Model | Method | Precision | | | | | | Granularity | |
|---|---|---|---|---|---|---|---|---|---|---|
| | | | 8-bit | 7-bit | 6-bit | 5-bit | 4-bit | 3-bit | Channel | Group |
| CIFAR-10 | ViT-Tiny | PTQ | 97.17 | 97.17 | 97.08 | 96.90 | 96.02 | 86.46 | 97.17 | 97.17 |
| | | Gen | **97.27** | **97.26** | **97.21** | **97.02** | **96.27** | **87.13** | **97.26** | **97.27** |
| | MLP-Mixer | PTQ | 96.56 | 96.54 | 96.54 | 96.44 | 96.24 | 92.67 | 96.57 | 96.56 |
| | | Gen | **96.61** | **96.64** | **96.61** | **96.49** | **96.33** | **93.27** | **96.64** | **96.61** |
| CIFAR-100 | ViT-Tiny | PTQ | 78.54 | 78.34 | 78.25 | 77.01 | 72.14 | **38.90** | 78.44 | 78.54 |
| | | Gen | **78.80** | **78.73** | **78.51** | **77.48** | **72.54** | 38.76 | **78.68** | **78.80** |
| | MLP-Mixer | PTQ | 96.49 | 96.52 | 96.45 | 96.45 | 96.01 | 93.31 | 96.50 | 96.49 |
| | | Gen | **96.64** | **96.70** | **96.59** | **96.64** | **96.34** | **93.71** | **96.65** | **96.64** |
| ImageNet-1K | ViT-Tiny | PTQ | 74.83 | 74.84 | 74.74 | 74.21 | 71.73 | **53.17** | 74.83 | 74.83 |
| | | Gen | **75.42** | **75.41** | **75.26** | **74.68** | **72.04** | 53.03 | **75.06** | **75.42** |
| | ViT-Small | PTQ | 81.42 | 81.42 | 81.33 | 81.06 | 80.19 | 72.53 | 81.41 | 81.42 |
| | | Gen | **81.50** | **81.49** | **81.45** | **81.34** | **80.34** | **72.70** | **81.51** | **81.50** |
| | ViT-Base | PTQ | 84.56 | 84.54 | 84.54 | 84.44 | 84.06 | 81.81 | **84.56** | 84.56 |
| | | Gen | **84.70** | **84.70** | **84.61** | **84.64** | **84.32** | **82.17** | 84.27 | **84.70** |

When employing the conditional parameter generation model to controllably generate quantized parameters, the process remains identical to the unconditional case except for the inclusion of additional control variables. By using our quantized parameter generation model, the entire quantization process requires no access to any target task data or additional fine-tuning, enabling an efficient and secure single-step quantization method.

## 4 EXPERIMENTS

In this section, we first provide a detailed description of the deployment details. We then employ our parameter generation model to produce quantized parameters of varying precision and granularity across different datasets and model architectures, and compare their performance with parameters obtained through PTQ or QAT. In addition, we design a series of ablation studies to validate the effectiveness of our proposed method.

### 4.1 EXPERIMENTAL SETUP

**Target Tasks.** We conduct experiments on benchmark datasets including ImageNet-1K (Deng et al., 2009), and CIFAR-10/100 (Krizhevsky, 2009), employing ViT (Dosovitskiy et al., 2020) series models and MLP-Mixer architectures from the timm[1] library as backbone models. To demonstrate the broad applicability of our proposed approach, we evaluate its effectiveness under both QAT and PTQ settings. For PTQ, we consider quantization precisions ranging from 3 to 8 bits, covering both channel-wise and group-wise granularities. For QAT, we evaluate 4-bit and 8-bit group-wise quantization.

**Parameter Collection.** All models utilize ImageNet1K pre-trained weights provided by the timm library. Under the PTQ setting, we apply one epoch of fine-tuning following quantization and extract 50 checkpoints to construct the training set for ImageNet-1K. For CIFAR-10, we freeze the feature extractor and fine-tune only the classification head, similarly collecting 50 checkpoints from the final

---

[1] https://github.com/huggingface/pytorch-image-models

training epoch. Under the QAT setting, all datasets undergo simulated quantization training, from which we collect 50 checkpoints from the final epoch to form the training set.

**Training and Inference Details.** For all models except ViT-Base, we set the token length to 8192. Due to the larger number of parameters in ViT-Base, a longer token length of 16384 is adopted to more effectively capture its parameter distribution. We adopt the learnable Mamba-v2 architecture as the default backbone for our parameter generation model. Training is performed with a batch size of 4 and 50,000 iteration. During inference, the model takes as input the full-precision parameter tokens, positional embeddings, and, if applicable, conditional control vectors. The generated quantized parameters are subsequently post-processed using rounding and clamping operations to ensure compliance with the target precision range. Their performance is then evaluated using simulated quantization (Loroch et al., 2017). The implementation details under different settings are provided in Appendix A.

## 4.2 MAIN RESULTS

**PTQ Parameter Generation.** PTQ is widely used in real-world deployments due to its simplicity and efficiency. Thus we conduct extensive comparisons across various precision and granularity settings under the PTQ configuration. For the precision experiments, we fix the granularity to the optimal group-wise setting. Conversely, for the granularity experiments, we adopt 8-bit quantization. As shown in Table 1, the quantized parameters generated by our model consistently achieve comparable or superior results across all configurations. These findings indicate that our method not only accurately models the affine transformation between full-precision and quantized parameters, but also effectively extracts task-relevant information from full-precision models, thereby mitigating quantization-induced performance degradation and outperforming traditional PTQ in many scenarios. We provide more experimental results of quantized parameter generation across various tasks in Appendix B.

**QAT Parameter Generation.** To evaluate the generalizability of our approach, we conducted performance comparisons on ImageNet-1K using ViT-Tiny under two widely adopted precision settings under QAT setting. The experimental results are summarized in Table 2. Even when the model is trained with simulated quantization-aware training, our quantized parameter generator accurately captures the parameter distribution, achieving comparable performance. Furthermore, we report the inference

Table 2: Performance comparison between QAT and our generation model. Our method could achieve comparable performance. 50× indicates that the metrics is computed based on 50 models.

| Method | 4-bit | | 8-bit | |
|---|---|---|---|---|
| | Acc (%) | Time (min) | Acc (%) | Time (min) |
| QAT | **72.23** | 1000 (50×) | 75.61 | 1000 (50×) |
| Ours | 71.78 | 300 (50×) | **75.73** | 300 (50×) |

time required by different methods. The notation "50×" indicates that all metrics are averaged over 50 sets of quantization parameters, reflecting realistic deployment scenarios. As also shown in Figure 1, our method exhibits a substantial efficiency advantage when multiple quantized models are deployed concurrently.

**Controllable Quantized Parameter Generation.** We further investigated controllable conditional quantization parameter generation to enable a truly one-for-all general-purpose model, which is fundamentally different from traditional quantization approaches. To this end, we designed two one-hot vectors representing the quantization conditions of precision and granularity, with each vector's length corresponding to the number of condition categories.

As presented in Table 3, we performed performance comparisons for the quantization of ViT-Tiny on ImageNet-1K. Despite substantial distributional differences among the target configurations, our generator was able to accurately produce appropriate quantization parameters based on the provided conditions. Furthermore, in contrast to the results shown in Table 1, condition-controlled generation model consistently resulted in improved performance across all configurations, demonstrating that our method effectively learns the general distribution of model parameters under various quantization schemes and produces more stable outputs.

Table 3: Performance of the controllable quantized parameter generation model. "P" and "G" denotes the precision and granularity conditions. When controlling for precision (P), granularity is fixed (resulting in NA values for granularity columns), and vice versa when controlling for granularity (G). Our proposed model achieves strong performance across all evaluated settings.

| Dataset | Method | Condition | Precision | | | | | | Granularity | |
|---|---|---|---|---|---|---|---|---|---|---|
| | | | 8-bit | 7-bit | 6-bit | 5-bit | 4-bit | 3-bit | Channel | Group |
| CIFAR-10 | PTQ | NA | 97.17 | 97.17 | 97.08 | 96.90 | 96.02 | 86.46 | 97.17 | 97.17 |
| | Gen | P | **97.24** | **97.23** | **97.19** | **96.97** | **96.20** | **86.85** | NA | NA |
| | | G | NA | NA | NA | NA | NA | NA | **97.21** | **97.27** |
| CIFAR100 | PTQ | NA | 78.54 | 78.34 | 78.25 | 77.01 | 72.14 | 38.90 | 78.44 | 78.54 |
| | Gen | P | **78.64** | **78.51** | **78.45** | **77.42** | **72.27** | **39.14** | NA | NA |
| | | G | NA | NA | NA | NA | NA | NA | **78.63** | **78.75** |
| ImageNet-1K | PTQ | NA | 74.83 | 74.84 | 74.74 | 74.21 | 71.73 | 53.17 | 74.83 | 74.83 |
| | Gen | P | **75.21** | **75.12** | **74.96** | **74.75** | **72.28** | **53.38** | NA | NA |
| | | G | NA | NA | NA | NA | NA | NA | **75.23** | **75.45** |

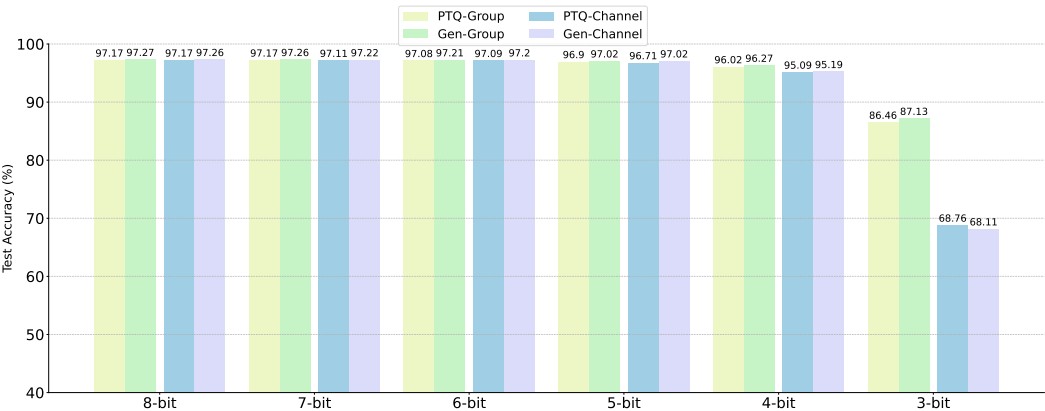

Figure 3: Extended performance results across all precision and granularity combinations. Our method demonstrates an effective and generalizable quantization framework across different quantization method and granularity.

### 4.3 ABLATION STUDIES AND ANALYSIS

**Combination of Precision and Granularity.** In above experiments, the default quantization precision for different granularity levels is set to 8-bit. To further evaluate the effectiveness and generalization ability of our method, we explored various combinations of precision and granularity under the same experimental conditions. As shown in Figure 3, we present a performance comparison of ViT-Tiny quantized on CIFAR-10.

It can be observed that across all quantization precision levels, the group-wise setting, due to its finer-grained partitioning, enables our method to focus primarily on capturing the overall parameter distribution. Consequently, it can leverage the full-precision parameter distribution for effective information compensation, consistently achieving superior performance. However, under the 3-bit and group-wise quantization setting, the parameter distribution becomes more constrained and exhibits finer variations. In this scenario, our method achieves the second-best performance, as the narrower distribution presents greater challenges for accurate approximation.

Table 4: Ablation study on the token size. The number refers to the length of each token. All the experiments are conducted with group-wise granularity.

| Method | 8-bit | 7-bit | 6-bit | 5-bit | 4-bit | 3-bit |
|---|---|---|---|---|---|---|
| Original | 84.56 | 84.54 | 84.54 | 84.44 | 84.06 | 81.81 |
| $Gen_{8192}$ | 83.92 | 58.25 | 54.76 | 59.04 | 55.32 | 56.56 |
| $Gen_{16384}$ | **84.70** | **84.70** | **84.61** | **84.64** | **84.32** | **82.17** |

Table 5: Ablation study on removing diffusion model.

| Method | Acc. |
|---|---|
| Original | 74.83 |
| + diffusion | fail |
| - diffusion | **75.42** |

Table 6: Ablation study on the padding value.

| Padding | Acc. |
|---|---|
| Original | 74.83 |
| 0 | 74.57 |
| 1 | 72.94 |
| Our | **75.42** |

**The manner of Token Size.**    As shown in Table 4, we quantized ViT-Base on ImageNet-1K and analyzed the impact of using input tokens of varying lengths in the parameter generation model. The results indicate that due to the larger number of parameters in ViT-Base, longer token sequences are necessary to effectively capture the dependencies among parameters. In contrast, for smaller models, we uniformly adopt a token length of 8192 to achieve better overall performance.

**The Influence of Diffusion Model.**    In contrast to existing parameter generation methods (Wang et al., 2025; Khan et al., 2025) that employ recurrent diffusion architectures, our goal is to model the transformation between full-precision parameters and their corresponding quantized versions using a generative model. Introducing a diffusion process into this framework often leads to uncontrollable distribution shifts, which are undesirable for the quantization task.

To investigate this issue, we integrated the diffusion architecture proposed in RPG into our model and conducted quantization experiments on ViT-Tiny using the ImageNet dataset. As shown Table 5, the inclusion of the diffusion model degraded the performance of our generator, as it failed to capture the parameter mapping effectively and thus could not learn meaningful representations. Furthermore, the diffusion process introduces considerable computational overhead during both training and inference, rendering it impractical for efficient deployment. We provid additional ablation study on recurrent generation model in Appendix C.

**The Effect of Padding Value.**    We further investigated the equally important parameter pre-processing step of filling blank positions in the input tokens, i.e., the last parameter tokens and quantization information tokens. In contrast to our default initialization strategy using Gaussian noise, we additionally explored two alternative padding methods: all-zeros and all-ones.

The quantization results of ViT-Tiny on ImageNet are presented in Table 6. It can be observed that zero-padding leads to a moderate performance decline, whereas one-padding causes a substantial degradation. We attribute this to the inherent sparsity of quantized parameters, which frequently include values close to zero. Padding with other constant values, such as ones, introduces a mismatch with the target distribution, thereby increasing optimization difficulty. In contrast, Gaussian noise initialization perturbs values around zero, offering a more suitable and optimizable starting point that aligns better with the true parameter distribution.

## 5    CONCLUSION

In this work, we propose a novel framework that extends parameter generation techniques to the domain of model quantization. In contrast to existing methods that primarily aim to regenerate parameters tailored to the same task objectives of trained parameters, our method offers a more practical and impactful application of parameter generation. By identifying the limitations of previous methods, we develop a dedicated parameter preprocessing technique optimized for the quantization process. Moreover, we remove the dependency on diffusion models, thereby substantially improving both training and inference efficiency. Our method supports conditional and controllable generation across multiple quantization precisions and schemes, enabling a unified one-for-all quantization paradigm. Extensive experiments show that our method can directly produce quantized model parameters with competitive performance through a single forward pass, without access to target task data or the need for fine-tuning. These results highlight the potential of our method as an efficient and scalable solution for model deployment.

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

# APPENDIX

## A IMPLEMENTATION DETAILS

Table 7: Detailed specifications of the different versions of the recurrent model. "Generated Param" denotes the number of target quantization parameters that the corresponding version of the generative model is designed to produce. The remaining hyperparameters correspond to those used in the Mamba layers.

| Version | Generated Param | 1st Dimension | 2nd Dimension | State Dimension | Conv Dimension | Expansion | Param Counts |
|---------|-----------------|---------------|---------------|-----------------|----------------|-----------|--------------|
| Base | 5M~50M | 8192 | 8192 | 128 | 4 | 2 | 1018M |
| Large | >50M | 12288 | 16384 | 128 | 4 | 2 | 3076M |

By default, we adopt a recurrent architecture consisting of two Mamba layers as the backbone. However, as the number of quantized parameters to be generated increases, a larger recurrent model is required to effectively capture the underlying dependencies among parameters. As shown in Table 7, we provide the detailed configurations of the two versions of our proposed recurrent model. The Large version is used for generating parameters of ViT-Base, while the Base version is employed in all other cases.

## B PERFORMANCE ON MORE TASKS

We validated the effectiveness of our proposed quantized parameter generation model across a diverse set of classification datasets and model architectures. To further evaluate the generalization capability of our approach, we extended our experiments to more complex tasks.

### B.1 QUANTIZED PARAMETER GENERATION ON DETECTION TASK

Table 8: Comparison of PTQ and our quantized parameter generation model on segmentation task. Our method still achieve competitive performance.

| Dataset | Model | Method | Precision | | | | | | Granularity | |
|---------|-------|--------|-----------|-----------|-----------|-----------|-----------|-----------|-------------|-------|
| | | | 8-bit | 7-bit | 6-bit | 5-bit | 4-bit | 3-bit | Channel | Group |
| ADE20K | UperNet | PTQ | **41.78** | **41.30** | **41.33** | **41.19** | **40.90** | **39.75** | 41.32 | **41.78** |
| | | Gen | 40.54 | 39.88 | 40.26 | 39.86 | 39.04 | 35.93 | **42.12** | 40.54 |

For the segmentation task, we quantized UperNet Xiao et al. (2018) which leverages Swin Transformer Liu et al. (2021) as the backbone on the ADE20K Zhou et al. (2017) under two settings: PTQ and direct parameter generation using our quantized parameter generation model. We adopt mean Intersection over Union (mIoU) as the evaluation metric. As shown in Table 8, although our method did not consistently outperform PTQ as it did in classification tasks, it still delivered competitive performance across all configurations. These results demonstrate the generalizability and effectiveness of our approach, while also highlighting the need for future research to further extend and optimize the method for more complex tasks.

### B.2 QUANTIZED PARAMETER GENERATION ON SEGMENTATION TASK

Table 9: Comparison of PTQ and our quantized parameter generation model on detection task. Our method achieve comparable and even better performance.

| Dataset | Model | Method | Precision | | | | | | Granularity | |
|---------|-------|--------|-----------|-----------|-----------|-----------|-----------|-----------|-------------|-------|
| | | | 8-bit | 7-bit | 6-bit | 5-bit | 4-bit | 3-bit | Channel | Group |
| COCO2017 | RT-DETR-V2 | PTQ | 45.7 | **45.7** | 45.7 | **45.5** | **45.3** | **37.6** | **45.7** | 45.7 |
| | | Gen | **45.7** | 45.6 | **45.9** | 45.4 | 44.1 | 31.6 | 45.2 | **45.7** |

For the object detection task, we conducted quantization experiments on RT-DETR-V2 Lv et al. (2024) using the COCO2017 Lin et al. (2014), we follow the official mean Average Precision (mAP) computation protocol provided by COCO, which samples 100 points on the precision-recall (PR) curve. Instead of using a fixed IoU threshold of 0.5, we compute the Average Precision (AP) at multiple IoU thresholds ranging from 0.5 to 0.95 with an interval of 0.05, and report the average of these AP values as the final result. As shown in Table 9, in contrast to the segmentation task, our approach achieved comparable or even superior performance under high-precision quantization settings. These results further validate the effectiveness and potential of directly generating quantized parameters.

However, under lower-precision settings such as 4-bit and 3-bit, our method exhibited a notable decline in performance. This observation suggests that, for more complex tasks, finer-grained constraints may be necessary during parameter generation. Addressing this challenge represents an important direction for future work.

## C   RETHINKING ON RECURRENT ARCHITECTURES

To further investigate the underlying mechanisms of our proposed quantized parameter generation model, we explore the use of different recurrent architectures as the backbone and compare their task performance as well as computational efficiency. The experimental results of quantized Vit-Tiny on ImageNet-1K are shown in Table 10, Mamba, which leverages the SSM technique, significantly outperforms both LSTM and Transformer in terms of time and memory efficiency.

Table 10: Ablation study on the architecture of recurrent model. In addition to significantly reducing computational and memory overhead, Mamba is the only architecture among those evaluated that effectively captures the distribution of model parameters.

| Architecture | Accuracy (%) | Time (h) | Memory (GB) |
|---|---|---|---|
| LSTM | 0.10 | 14.5 | 4.03 |
| Transformer | 0.18 | 3.78 | 3.53 |
| Mamba | **75.42** | **3.67** | **3.05** |

In addition, we observe that Mamba is uniquely capable of effectively capturing the intrinsic relationships among parameters and accurately modeling the quantization affine transformation. In contrast to RPG Wang et al. (2025) and ORAL Khan et al. (2025), where replacing the recurrent module still preserves effectiveness, our approach provides a more in-depth exploration of the structural properties of recurrent architectures and incorporates task-specific adaptations designed specifically for model quantization.

