# OpenReview forum: "QuantGen: Parameter Generation for Controllable Model Quantization"
_ICLR.cc/2026/Conference — Submitted to ICLR 2026_

### Official Review · Reviewer_rtmi · 2025-10-24

**Soundness:** 2
**Presentation:** 3
**Contribution:** 2
**Rating:** 4
**Confidence:** 4

**Summary:**

The paper introduces *QuantGen*, a recurrent parameter generator that directly maps full-precision weights to quantized weights in a single step with claims that it does so without access to target task data. The core design of QuantGen treats weights as tokens, and injects quantization metadata $(S,Z)$ into the token stream, and trains with an MSE loss to regress PTQ/QAT-produced ground truths. The paper shows that the model supports controllable precision and granularity via one-hot condition embeddings. The paper conducts comprehensive experiments on image models, which show modest gains over PTQ with near-QAT results.

**Strengths:**

- The authors formulate *parameter generation* as an interesting alternative to quantization, showing better performance than PTQ and less computational resources than QAT.
- One-hot condition embeddings for precision and granularity make the generator generalizable, with consistent gains on multiple models.
- The writing is clear and easy to follow.

**Weaknesses:**

1. The core equations mentioned in the paper are the usual affine quantization and dequantization. The novelty is mainly system-level (tokenization + recurrent generator + simple mse loss) which makes the contribution feel incremental compared to methods to push PTQ/QAT.
2. To my understanding, the generator must be trained with PTQ/QAT ground-truths per architecture–dataset configuration, as it learns only within that distribution of checkpoints. This design limits the “data-free” and “amortized” efficiency claims since each new architecture or dataset still requires collecting PTQ/QAT checkpoints and retraining the generator from scratch. The method thus behaves more like a model-specific quantization accelerator than a general-purpose quantizer.
3. As the generator is dependent on PTQ/QAT targets rather than optimizing downstream metrics directly. This likely upper bounds the performance gain in cases where QAT is suboptimal (like in large language models), which makes QuantGen an approximation of PTQ at best.

**Questions:**

1. Instead of regressing to PTQ/QAT parameters, did you try optimizing a proxy like layer-wise calibration loss or output-space MSE directly from fp-32 to quantized (still data-free via synthetic stats)? If not, what blocked it?
2. Is the generator trained and reused across different architectures and datasets, or does it have to be retrained with PTQ/QAT ground truths for each new model architecture + dataset pair? If retraining is required, can you quantify this cost and clarify how it affects the claimed amortization and scalability benefits? What exactly is shared and what is retrained?
3. Is it possible to remove the QAT dependency by training the generator only from PTQ (or data-free) teacher tokens and still match QAT accuracy? That would strengthen the “no data” claim.
4. As it is now, the main advantage of QuantGen seems to come from the ability to quantize large numbers of the *same* model architecture on a *single* task/domain. What are some practical use-cases of this approach, which cannot be mitigated by PTQ/QAT?

---

### Official Review · Reviewer_rmco · 2025-10-30

**Soundness:** 3
**Presentation:** 3
**Contribution:** 3
**Rating:** 4
**Confidence:** 3

**Summary:**

This paper proposes an efficient method for parameter generation in model quantization. It builds upon recent work, RPG, which combines a recurrent neural network learning approach with a diffusion model. The paper addresses the challenges of current quantization methods by proposing a conditionally controllable and flexible process. The paper focuses on the recurrent neural network part of RPG, i.e. using Mamba for generating quantized parameter tokens. This approach avoids the diffusion-based model, resulting in a more efficient method. The output of the recurrent model is then compared to the target parameter tokens, which are quantized by current quantization techniques, post-training quantization (PTQ) and quantization-aware training (QAT). The proposed method is evaluated on CIFAR-10/100 and ImageNet-1k using two different transformer models, VIT and MLP-Mixer. The comparison to PTQ and QAT demonstrates improvements in most quantization rates.

**Strengths:**

- This paper presents good motivation why model quantization is important
- This paper presents an important approach of parameter generation, elevating efficiency and more generalizability for actual applications
- The paper also shows that including the additional overhead of a diffusion model is not needed to generate strong parameters

**Weaknesses:**

- This approach is somehow limited in novelty because Mamba is already known in the context of parameter generation.
- While the topic is important, it relies on a generated dataset (here 50 checkpoints) of fully trained networks and their quantized counterparts. This raises the question of its applicability to new networks.
- The claim that the performance in line 345 is superior feels a bit too strong here.
- How do the generated parameters differ from the trained ones? This comparison helps assess the method’s ability to generate novel parameters rather than memorising the trained ones.
- Diffusion models are primarily used to generate a diverse range of data points, so it’s concerning that it fails in this context. Any explanation for the model’s failure in Table 5 should be provided.
- The training data is based on the same network topology but trained differently. This suggests that each network topology is handled individually, which seems contradictory to the claim that the proposed method is generalisable to a wide range of networks in lines 096-097.

*Minor*:
- Text in figure 3 is too small

**Questions:**

- The difference between PTQ and Gen is quite small, so what is the computational difference? For Gen, one first needs to generate a complete dataset to also perform post-training quantization. What is the trade-off in terms of compute?
- Is it correct, that at inference time the model uses full-precision networks and aims to generate quanzited versions, i.e. the overall goal is to quantize already trained networks?
- Could we generate quantized parameters for untrained networks?
- So far, the paper has shown experiments on two different networks individually. How about generating parameters for unseen, novel networks? Alternatively, could we train on MLP-Mixer and generate parameters for VIT?
- Is the simulated quantizations also a post-processing step to evaluate the performance of generated parameters? What is the overhead involved for that post-processing step then (Line 335ff)?

---

### Official Review · Reviewer_EscZ · 2025-10-31

**Soundness:** 2
**Presentation:** 1
**Contribution:** 2
**Rating:** 2
**Confidence:** 4

**Summary:**

This paper proposes QuantGen, a recurrent-based framework for generating quantized model parameters directly from full-precision counterparts, bypassing traditional quantization pipelines that require retraining or calibration data. The method emphasizes controllability across precision levels and quantization schemes, while maintaining performance parity with conventional techniques. Experiments validate its efficiency, generalization, and privacy-preserving advantages.

**Strengths:**

1. The framework supports dynamic adjustments to precision (e.g., 8-bit to 3-bit) and quantization granularity (channel vs. group), validated through conditional generation experiments (Table 3). This aligns with real-world deployment needs for hardware compatibility.


2. Experiments across ImageNet, CIFAR datasets demonstrate competitive performance. For example, on ImageNet-1K with ViT-Tiny, QuantGen achieves 75.73% accuracy at 8-bit precision, marginally outperforming QAT (75.61%).

**Weaknesses:**

1. The proposed method builds upon conventional quantization models but incorporates additional data processing and training steps. However, it only achieves marginal improvement in accuracy over standard quantization techniques. What distinct advantages does this method offer compared to existing quantization approaches?

2. The method’s tokenization strategy (e.g., token length 16,384 for ViT-Base) may struggle with larger architectures like ViT-Large or modern LLMs. The computational trade-offs for billion-parameter models remain unaddressed.

3. Evaluations focus on classification and detection, but segmentation results (e.g., UperNet on ADE20K) show inconsistent gains (Table 8). The generalizability to generative tasks (e.g., diffusion models) is untested.

**Questions:**

Please see the weaknesses.

---

### Official Review · Reviewer_dbQd · 2025-11-02

**Soundness:** 2
**Presentation:** 3
**Contribution:** 2
**Rating:** 4
**Confidence:** 4

**Summary:**

This paper focuses on model quantization problems and proposes a new framework for data-free and direct weight generation. The proposed method outperforms QAT in classification tasks and other tasks (in appendix).

**Strengths:**

The paper is clearly written, with benchmarks with classification tasks and other tasks (in appendix). The proposed approach is fast and easy to use in practice since it does not require any data and retraining.

**Weaknesses:**

In this paper, QUANTGEN is compared to QAT with (a kind of limited) improvement. It would be better to expand the experimental section to induce more comparisons, which might include some recent advanced QAT result like https://arxiv.org/pdf/2407.11062

Considering the novelty and contribution, I lean towards rejection.

**Questions:**

1. Note that QUANTGEN fails or degrades in low-bit or complex-task settings. I would suggest that the authors provide more insight and analysis into the reasons behind this degradation.
2. While the major benefit of QUANTGEN is its ability to operate without task or data awareness, I am concerned that this approach might miss essential data-driven subtleties that affect end-task accuracy, especially in less standard network architectures. Do the authors believe that the lack of task or data awareness could limit QUANTGEN’s effectiveness in such cases?

---

### Meta-Review · Area_Chair_hNa9 · 2025-12-25

**Summary:**

The reviewers agree that the paper presents a clear and technically sound framework for data-free parameter generation for quantization.

Key concerns include:
* limited novelty beyond existing parameter-generation and quantization pipelines,
* marginal performance gains over strong PTQ/QAT baselines,
* unclear advantages relative to the additional training overhead.

No concerns were addressed, as no rebuttal was provided.

**Reviewer Concerns:**

None (no rebuttal).

**Reviewer Scores:**

Based on the absence of a rebuttal, the likely final scores remain: 4 / 2 / 4 / 4.

---

### Decision · Program_Chairs · 2026-01-26

Reject